# Mitigation Potential of Herbal Extracts and Constituent Bioactive Compounds on *Salmonella* in Meat-Type Poultry

**DOI:** 10.3390/ani14071087

**Published:** 2024-04-03

**Authors:** Oluteru E. Orimaye, Deji A. Ekunseitan, Paul C. Omaliko, Yewande O. Fasina

**Affiliations:** Animal Sciences Department, North Carolina A&T State University, Greensboro, NC 27411, USA; oeorimaye@aggies.ncat.edu (O.E.O.); daekunseitan@ncat.edu (D.A.E.);

**Keywords:** herbal extracts, poultry, *Salmonella*, phytobiotics, bioactive compounds, antimicrobials

## Abstract

**Simple Summary:**

Poultry is one of the important sectors in the livestock industry and has grown tremendously over the recent few decades. The poultry industry is faced with several challenges, but the key amongst them is disease colonization and antimicrobial residue (AMR). *Salmonella* is one of the major diseases of chicken that causes salmonellosis in humans. Many plant sources and their extracts were evaluated. This review aims to provide an overview of herbal extracts that can be employed to manage and control *Salmonella* infection and serve as an alternative to antibiotics in meat-type poultry production.

**Abstract:**

Herbal extracts have been widely evaluated in poultry production for their beneficial effects and potential substitute for antibiotics, which contribute to AMR and risks to human health through the consumption of infected meat. Salmonellosis is a systemic infection caused by *Salmonella*, an intracellular bacterium with the ability to cause systemic infections with significant implications for both the health and safety of farmers and consumers. The excessive use of antibiotics has escalated the incidence of antibiotic resistance bacteria in the poultry and livestock industry, highlighting the urgent need for alternatives especially in meat-type poultry. Both in vivo usage and in vitro studies of bioactive compounds from herbal extracts have demonstrated the effective antimicrobial activities against pathogenic bacteria, showing promise in managing *Salmonella* infections and enhancing poultry performance. Phytobiotic feed additives have shown promising results in improving poultry output due to their pharmacological properties, such as stimulating consumption, and enhancing antioxidant properties and preventing the increasing antimicrobial resistance threats. Despite potential for synergistic effects from plant-derived compounds, a further investigation into is essential to fully understand their role and mechanisms of action, for developing effective delivery systems, and for assessing environmental sustainability in controlling *Salmonella* in poultry production.

## 1. Introduction

Salmonellosis is a prevalent infection that significantly impacts commercial poultry operations, resulting in substantial production losses and presenting a public health risk. *Salmonella* is a rod-like-shaped Gram-negative bacteria that is capable of surviving with or without oxygen and is the primary cause of inflammation in the gastrointestinal tract, resulting in gastroenteritis. The most prevalent cause of gastroenteritis in people and animals is *Salmonella enterica serovar* Typhimurium, leading to acute gastroenteritis and bacteremia [1]. In vitro infection models have shown that *Salmonella* Typhimurium can induce a gradual reduction in transepithelial electrical resistance, modulate the tight junction (TJ) protein expression on epithelial cells of the intestine, impair the functionality of the digestive tract, and promote microorganism translocation [1]. The reduction in *Salmonella* colonization in the intestinal tract of meat-type poultry safeguards the structural integrity of the gut barrier and reduces potentials for carcass contamination during slaughter [1]. *Salmonella* remains the most important pathogen in terms of food safety on a global scale, and it is primarily transmitted through poultry. For so long, antibiotics have served as the primary approach to manage and control *Salmonella* infection in poultry production. In any case, the emergence of resistance to antibiotics among pathogenic bacteria has become a global discourse on the need to restrict antibiotic usage in animal agriculture [1]. In order to control *Salmonella* infection, various prophylactic strategies have been implemented, including increased hygiene standards, antibiotic use, probiotics, genetically selected chicken lines, and products from competitive exclusion, to enhance the advancement of the *Salmonella* vaccine and boost our immune response [1,2]. The persistent challenge of the disease and the antimicrobial resistance (AMR) problem has resulted in the search of alternatives, one of such is the use of herbal extracts from phytobiotics. However, phytobiotic feed additives (PFA) have shown promising results in terms of poultry output and have been used due to their pharmacological properties, such as stimulating consumption, enhancing antioxidant capacity, augmenting endogenous enzyme secretion, and imparting antibacterial properties. Research has also shown that herbal extracts alter the permeability and fluidity of the cell membrane, leading to an increase in nutrient absorption [3]. Additionally, studies have documented that the antioxidant capacity of meat-type poultry birds is enhanced by different herbal extracts, thereby enhancing the shelf-life of their meat and products. Juskiewicz et al. [3] found that the fatty acid composition of meat-type poultry was altered by bioactive components of herbal extracts; however, inconsistent outcomes regarding the supplementation of poultry feed with herbal extracts have been documented in recent studies [4], which may be attributed to factors such as the bacterial challenge employed, the timing of the challenge, or the specific strain of bacteria used in the investigation [3]. Additionally, Ibrahim et al. [5] noted that incorporating thymol into poultry diets can improve BWG, feed conversion rate, and regulated FI in broilers with *Salmonella* Typhimurium.

This review explores the potential of herbal extracts to mitigate intestinal *Salmonella* in meat-type poultry birds.

## 2. Prevalence of *Salmonella* in Meat-Type Poultry and Consequences on Human Health

Salmonellosis is a major problem in the poultry industry; it contributes to serious challenges and causes economic losses ranging up to more than USD 4 billion every year with regard to poultry farmers in the USA alone. The most common *Salmonella* serotypes found in poultry are Heidelberg, Kentucky, and Typhimurium [6]. Between 1999 and 2008, *Salmonella* caused 36,490 associated illnesses and 1335 foodborne outbreaks [5]. Poultry was responsible for a greater proportion of these outbreaks, with 35% of reported epidemics attributed to a single food source. *Salmonella* infection is sporadic, with reported cases exceeding instances connected to the surge by a factor of more than fifteen to one [5]. *S. enteritidis* and *S. typhimurium* had the greatest frequently documented serotype cases for human beings [7,8]. *Salmonella* Kentucky was identified in carcass surveillance programs with the highest frequency [7]. *Salmonella* pathogenicity has also been linked to exposure to poultry, with eggshells, live birds, and poultry products that have been processed implicated in ten out of twenty-five outbreaks from 2006 to the present [7]. In the poultry industry, *Salmonella* infections can infiltrate poultry processing through multiple routes. Contamination often begins at the farm level, where live birds can carry bacteria. During processing, improper hygiene practices, equipment, and the environment contribute to the spread. Cross-contamination on slaughter slabs can occur through contact with contaminated surfaces, tools, or from workers’ hands. Inadequate sanitation measures may allow the persistence and transmission of *Salmonella*. For a long time, antibiotics have been employed in controlling pathogenic bacteria, such as *Salmonella* species, and as a growth promoter in the poultry sector. However, the spread of antibiotic resistance genes has necessitated the need to look for alternative to antibiotic usage in meat-type poultry production. Effective control measures, including stringent hygiene protocols, proper equipment cleaning, and workers’ training, are also essential to minimize the risk of *Salmonella* infection from farms to the slaughter slabs in the poultry industry [6].

### 2.1. Salmonella Prevalence in Broiler Chickens

Identifying *Salmonella* serovars in poultry, especially chicken samples, is a serious public health concern since they may cause various illnesses both in humans and poultry [9]. Globally, about 93.8 million cases of non-typhoidal *Salmonella* (NTS) occur annually, resulting in 155,000 deaths and 82,694 confirmed cases, making it the second most frequent zoonotic disease since 2013 to date [10]. *Salmonella* contamination in human food has become a public health concern over time [11]. It is a major cause of diarrhea, the second most prevalent food-transmitted symptoms in both EU and US, and the third highest cause of mortality from foodborne infections globally. *Salmonella* frequency in chicken and poultry products varies between fresh markets and supermarkets [11]. However, the contamination or transfer of salmonellosis usually occur primarily at the farm level, during transportation, at the processing unit, and throughout the production chain, which makes it more difficult to control [12].

### 2.2. Salmonella Prevalence in Turkey

Since December 2017 to May 2019, various government agencies, including health department officials, the CDC, FDA, FSIS, APHIS, and NTF, investigated an outbreak of 356 cases of *Salmonella* reading infection linked to turkey consumption. The epidemic was associated with human and pet food items containing turkey, as well as with turkey production and processing settings. A single death was recorded in California. The strain of S. readings responsible for the spread was found in several turkey products and live turkeys. Unlike most previous instances of food-related illnesses that can be traced back to a particular brand or source of product, the available information indicates that this spread was connected to various marketers of turkey products and that the outbreak strain was widely distributed among the turkey business. The inquiry prompted two voluntary withdrawals of turkey goods controlled by the FSIS for human consumption and three voluntary recalls of turkey pet food regulated by the FDA. However, it is important to note that these recalled items did not explain all of the recorded cases of sickness in humans during this epidemic. The collaboration between the government and industry in responding to this *Salmonella* strain, with the aim of protecting consumers, has served as a benchmark for future investigations involving government–industry cooperation. The NTF has issued guidelines outlining the most effective methods for minimizing the presence of *Salmonella* in turkeys and turkey products in light of this epidemic. Several multitudes of individuals have developed illnesses as a result of consuming turkey products that have been tainted with *Salmonella* spp. [13]. It was also noted by the CDC in 2019 that an additional 63 individuals contracted the illness, resulting in a cumulative total of 279 cases throughout 41 states and the District of Columbia. Out of these instances, 107 individuals had such severe illnesses that they required hospitalization. A death was documented in California in relation to a *Salmonella* epidemic. Multiple outbreaks of *Salmonella* strains have been reported in different live turkeys, including ground turkey and turkey patties, as well as the whole turkey business. *Salmonella* infection manifests via symptoms such as diarrhea, fever, and stomach cramps. These symptoms usually start within a timeframe of 12 to 72 h after consuming food that has been infected with the bacterium [13].

### 2.3. Factors Influencing Salmonella Colonization in Meat-Type Poultry Birds

*Salmonella* colonization in chicken is influenced by numerous factors, and understanding these factors is crucial for developing effective strategies to prevent and control *Salmonella* contamination in poultry. Most of these factors include age, and physiological and environmental stressors. Stress is a crucial factor since it weakens birds’ immune systems, making them more vulnerable to *Salmonella* due to overcrowding and environmental changes [14]. The likelihood of *Salmonella* colonization is increased by malnutrition, concomitant illnesses, and immunosuppressive conditions. Likewise, the quality of feed and water is also a crucial factor since polluted supplies may introduce and sustain the bacteria in the flock [14]. The incidence of *Salmonella* is also influenced by housing circumstances, namely ventilation, temperature regulation, and hygiene, and the interaction with nature and other animals poses extra hazards, since wild birds and rodents may potentially transmit diseases. Additionally, some of these factors are summarized below.

#### 2.3.1. Gut Microbiota

The gut is a vital organ in poultry birds responsible for breaking down food into smaller particles and transporting them to the body’s cells for energy production. The poultry gastrointestinal system is similar to mammals but with avian-specific structures like the crop, gizzard, ceca, and cloaca. The esophagus connects their beaks to their stomachs, where the avian GI system originates [14]. Poultry birds’ stomachs consist of proventriculus and ventriculus, which function as a microbiological barrier due to their low pH values. The small intestine consists of the duodenum, jejunum, and ileum, which facilitate digestion, produce bicarbonate and digestive enzymes, and absorb nutrients. Lactobacillus is the main bacterial species found in all three parts of the small intestine [14]. The ceca, two blind pouches connected to the small intestine, play a crucial role in nutrient fermentation. The cecum microbiota is more diverse and richer than the small intestine microbiome, including *Enterococcus*, coliforms, *Lactobacillus*, and yeast. Chickens’ intestines are generally sterile in the early days after hatching while the gut microbiota develops [14]. *Salmonella* thrives in this environment, and exposure to *Salmonella* reduces microbial diversity, leading to opportunistic infections in the stomach. The gut and lymphoid tissue (GALT) protects against harmful intestinal bacteria species [14].

#### 2.3.2. Management

Meat-type poultry birds may become infected with *Salmonella* via a direct contact with other disease-infested birds. This can occur during transit to the farm or introduction of new birds into an established flock. Food or water contaminated with infected bird excrement may also spread *Salmonella*, while unsanitary living conditions can raise the risk of *Salmonella* infection in hens. Overcrowding, unhealthy environments, and poor waste management may all aid in the spread of germs [14]. Many farms adhere to strict cleanliness and disease control practices to help prevent *Salmonella* from contaminating poultry meat and products. This involves frequently serological testing birds for *Salmonella*, managing their living circumstances, and adhering to basic food safety practices throughout processing and handling. Regularly removing manure, bedding, and other waste items from the coop and replacing them with new while exercising excellent hygiene using protective gear and equipment as needed is important [14], as dry materials may help to prevent the occurrence of *Salmonella*.

#### 2.3.3. Age

Age is a crucial factor, with *Salmonella* colonization being most prevalent in newly hatched chicks due to immature intestinal microbes inside the alimentary region. Chicks display symptoms sooner after being exposed to *Salmonella* due to their increased susceptibility to infection. Even in quantities of cells (lower than ten), *Salmonella* will easily spread the disease to young chickens that are newly hatched. However, the likelihood of young birds becoming infected with *Salmonella* diminishes as they grow older [14]. Thirty-eight percent of intra cloacal inoculated 1-day-old chickens were found to be colonizable with as few as two *Salmonella* cells [14]. The number of cells needed to achieve a colonizing dose of 50 via oral and intra cloacal inoculation was one hundred times smaller than that of chicks aged three days that had been fed [14]. There is susceptibility of young chicks (1 day to 2 weeks) to *Salmonella* infection, persisting until they reach maturity, resulting in malabsorption, depressed growth rate, inefficient feed utilization, and mortality [15].

#### 2.3.4. Strain

Different *Salmonella* strains have different incubation durations and clinical presentations in chickens. Strain heterogeneity and genetic variables contribute to the differing levels of virulence seen in various *Salmonella* strains [16]. The sensitivity or resistance to *Salmonella* infection differs across different chicken breeds due to genetic factors [17]. Strain variability in *Salmonella* refers to the genetic diversity among different strains of the bacteria. The impact of strain variability on *Salmonella* colonization in broiler chickens is complex and can influence the ability of *Salmonella* to establish and persist within the chicken population. Different *Salmonella* strains may exhibit varying degrees of adaptation to poultry hosts. Some strains may have specific genetic traits that enhance their ability to colonize the gastrointestinal tract of broiler chickens, while others may be less adapted. Strains with specific virulence factors or genetic attributes may have an advantage in overcoming host defenses and successfully colonizing the chicken’s gastrointestinal tract [17]. The ability of *Salmonella* strains to spread within and between flocks may vary. Some strains may have a higher transmission potential, leading to a more widespread and persistent presence in the broiler chicken population. Certain *Salmonella* strains may carry antibiotic resistance genes, allowing them to withstand the effects of antibiotics commonly used in poultry farming. Antibiotic-resistant strains may persist longer in the presence of antibiotics, complicating control measures and potentially leading to more prolonged colonization. Variability in *Salmonella* strains can influence the host’s immune response. Some strains may trigger a stronger or weaker immune response in broiler chickens, impacting the ability of the host to clear the infection. This interaction between strain variability and the host immune system plays a crucial role in *Salmonella* colonization dynamics. These interactions can influence the competition for resources and may impact the overall colonization success of specific *Salmonella* strains. Understanding the strain-specific factors that contribute to *Salmonella* colonization is essential for developing targeted control strategies. This may include the development of vaccines specific to prevalent strains, the implementation of biosecurity measures, and tailored management practices to mitigate the impact of highly colonizing strains [17].

### 2.4. Detrimental Effects of Salmonella Carriage in Poultry on Human Health

*Salmonella* infection is the most common cause of foodborne illnesses globally, infecting the gastrointestinal system and causing diarrhea, nausea, and cramps among human beings. However, the CDC reports that roughly 1.35 million outbreaks and 420 deaths occur each year in the United States [18]. Poultry is the primary reservoir for *Salmonella* infections with non-typhoidal serotypes among the animals that produce food. Non-typhoidal *Salmonella* connected with meat-type poultry birds causes major health challenges and economic stress for society, with an anticipated yearly cost of USD 2.79 billion. This situation is developing as the worldwide demand for ready-to-eat food items increases [18]. The listed examples below stem from a direct effect of *Salmonella* on human health.

#### 2.4.1. Antibiotic Resistance

Animal-derived goods have negative implications on One Health, including increased greenhouse gas emissions, contamination of drinkable water, pollution of the environment, the spread of antimicrobial medication resistance, and the advent of zoonotic illnesses. Antibiotics transformed medicine by combating bacterial infections, increasing lifespans for humans and animals, controlling infectious diseases, reducing morbidity and mortality, and improving food safety [19,20]. The widespread usage of these chemicals has led to the emergence and spread of multidrug-resistant (MDR) bacteria, causing a worldwide alarm [21]. In 2000, the World Health Organization identified antimicrobial drug resistance (AMR) as a worldwide public health risk. However, strategies for controlling and mitigating these pressures were urgently sought for then [22]. In 2015, the World Health Assembly adopted a global action plan to address AMR. The plan emphasizes the importance of the One Health approach that involves a collaboration among various sectors, including doctors, farmers, economists, environmentalists, and the public. Antibiotics are often used in livestock and poultry production for medicinal, preventive, and growth promotion purposes. However, improper use may lead to the formation of microbial reservoirs with AMR determinants in farm animals, not excluding poultry. AMR presents a significant danger to treating major bacterial infections in people, resulting in greater medical expenses, longer hospital stays, and increased death rates. AGP promotes gut bacteria selection, reduces nutritional competition, and boosts animal growth rates. AGP usage has aided the development and spread of AMR in the gut microbiota, forcing several governments to prohibit its usage in poultry and animal agriculture [23]. Drug resistance may spread across the food chain by direct or indirect interaction between multiple parties and environments, both of which are considered mode of spread for zoonotic illnesses. Direct exposure happens when people come into contact with antibiotic-resistant bacteria found in animals or their natural products, such as urine, feces, blood, and saliva. Professional employees, including veterinarians, farmers, slaughterhouse staff, and those handling food, as well as others who come into contact with them, are more likely to become colonized or infected with resistant strains. Many antimicrobials remain active even after being expelled in urine and feces, since they are not completely destroyed by animals or human beings. *Salmonella* levels on retail poultry products routinely exceed the 9.8% threshold for plant processing. A study conducted by Mazengia et al. [24] on meat-type poultry products at retail stores discovered that an average rate of 11.3% salmonellosis incidence remains high at 15.3 cases per 100,000, exceeding the CDC’s 2030 healthy people target of 11.5 cases per 100,000 populations (U.S. DHHS, 2020) [25]. Despite an ongoing aim to decrease this prevalence over the recent two decades (CDC, 2017) [26], there has been no significant decrease.

#### 2.4.2. Gastroenteritis

In poultry, gastroenteritis is an inflammation of the stomach lining as well as the small and large intestines. This actually affects the intestinal linings of the birds.

##### Gastroenteritis as a Foodborne Illness and its General Symptom in Human

*Salmonella* spp. is the main cause of human salmonellosis in the world, with contaminated eggs and raw poultry meat as the major routes of infection in spite of the successful implementation of *Salmonella* control measures in food/animal production in developed and industrialized countries. Non-typhoidal *Salmonella* (NTS) is one of the major zoonotic foodborne pathogens and constitute a menace to health problems worldwide [15]. *Salmonella* infections in humans by NTS is usually characterized by a self-limiting gastroenteritis syndrome and symptoms such as diarrhea, discomfort in the abdomen, fever, anorexia, and nausea. The persistence of the bacteria in the intestinal tract of food-producing animals contributes to its contamination of diverse food materials from poultry industry [27], leading large outbreaks of foodborne diseases and a corresponding increase in the cost of treatment.

##### Consumption of Improperly Cooked and Contaminated Poultry Products

Symptoms of salmonellosis may vary clinically from ordinary symptoms of other disease-causing agents that include diarrhea, abdominal cramps, and fever, to enteric fevers like typhoid fever, which are life-threatening febrile systemic illnesses that need immediate antibiotic treatment. Salmonellosis is a zoonotic disease with a large animal reservoir; hence, contaminated food is the primary mechanism of transmission for non-typhoidal *Salmonella*. Chickens and turkeys are the most frequent animal reservoirs, although dozens of other domestic and wild species also host these germs. *Salmonella* may live in meats and animal products that have not been completely cooked. The extent of the issue of *Salmonella* gastroenteritis is indicated by the following recent *Salmonella* yields: 41% of turkeys were investigated in California, 50% of hens were raised in Massachusetts, and 21% of commercial frozen egg whites were tested in Spokane, Washington [28]. Typhoid fever, along with other enteric fevers, is predominantly transmitted from human to human because these pathogens lack a major reservoir in animals. Exposure to human waste products is the primary mechanism of dissemination, and polluted water is the most common mode of transportation. Occasionally, contaminated food that is typically touched by someone infected with *S. typhimurium* may serve as the carrier. Plasmid DNA fingerprinting and bacterium phage lysotyping of *Salmonella* isolates are strong epidemiologic methods for investigating salmonellosis outbreaks and tracking the organisms’ dissemination in the environment. There are variables that influence the epidemiology of typhoid fever and non-typhoidal salmonellosis, in which there is an asymptomatic human carrier state for any type of illness. Approximately 3% of those infected with *S. typhi* and 0.1% with non-typhoidal *Salmonella* develop chronic infections [28].

##### Statistical Evidence of the Impact on Human Health

The majority of human infections are caused by just a few non-typhoidal *Salmonella* (NTSs), and out of these, *S. enteritidis* and *S. typhimurium* are regarded as the most significant serovars with the largest effect on public health, accounting for more than 70% of human infections. The majority of serovars are non-pathogenic in animals but highly pathogenic in humans. *S. enteritidis* and *S. typhimurium* are the most common serovars causing human illness in the European Union (EU). The European Food Safety Authority [13] has reported an increase in the frequency and severity of human infections caused by *S. typhimurium*. Domestic and wild animals, including poultry, swine, and cattle, can host *Salmonella* and act as reservoirs. The slaughter process may potentially be a source of infection, particularly if proper hygiene standards are not followed. *Salmonella* found in the intestinal fluids of carrier animals may contaminate the slaughter process, including vehicles, lairages, slaughter lines, and quartering [29]. Foodborne outbreaks have been linked to different sources from mixed feeds. Efforts to minimize *Salmonella* transmission by food and other pathways must be conducted using a single health strategy. Salmonellosis prevention involves a comprehensive effort at all stages of the food chain, including farming, manufacture, distribution, and consumption. However, the EU database indicated that there were 4088 foodborne outbreaks (FBOs) in 2021. In these FBOs, there were 33,813 human cases, 2560 hospitalizations, and 33 deaths. Between 2016 and 2021, these values decreased somewhat, except for the death variable, which increased due to high mortality rates in 2019 and 2020 in human listeriosis cases. The FBO-dashboard interactive tool displays outbreaks and cases per 100,000 people by nation, as well as the number of human cases and disease agents. It also ranks outbreaks based on food vehicle and area of exposure. According to the most current EU One Health Zoonoses Report [30], *Salmonella* spp. is the second most frequent zoonotic pathogen behind *Campylobacter*, with both causing gastrointestinal diseases in humans. Between 2017 and 2021, there were 60,050 confirmed cases of salmonellosis, with an EU notification rate of 15.7 per 100,000 individuals. The trend remained constant. There were 11,790 hospitalizations (45.0% of outbreak-related illnesses) and 71 reported deaths. According to EFSA statistics, it is the leading cause of foodborne outbreaks, accounting for 773 human cases (20.8%), 1123 hospitalizations, and 1 death. The main five serovars causing human illnesses are from strains *S. enteritidis* and *S. typhimurium*. According to EFSA data, the majority of *Salmonella* spp. isolates originate from broiler production (55.7%), followed by turkeys (12.9%), pigs (7.6%), and laying hens (6.0%). The data were collected from chicken populations covered by the *Salmonella* National Control Program (SNCP) [30]. *S. infantis* was found to be entirely connected with broiler sources (95.2%), whereas *S. enteritidis* was mostly related to broiler flocks and meat (70.0%) and laying flocks and eggs. In 2021, there was a considerable rise in *Salmonella* prevalence among breeding turkey flocks. In contrast, flock prevalence patterns for targeting *Salmonella* serovars have remained steady across all poultry populations in recent years [30]. A summary of salmonella infection pathways is depicted in Figure 1.

## 3. Herbal Extracts and Constituent Bioactive Compounds as Alternatives to Antibiotics for *Salmonella* Control in Meat-Type Poultry

### 3.1. Importance of Conducting Research in Identifying Effective Alternative to Antibiotics

Antibiotics have played an important part in the expansion and profitability of the poultry industry, but there have been various worries that their usage has resulted in the development of antimicrobial resistance (AMR), which poses a potential hazard to the health of people [31]. However, there are currently differing perspectives on the transmission of antibiotic resistance genes through animal-to-human infections. Several investigations found a relationship between the use of subtherapeutic antibiotics and the establishment of AMR in microflora [31]. It is anticipated that by 2050, roughly 10 million deaths will occur annually, resulting in a total loss of USD 100 trillion to the world economy if substantial steps are not taken to arrest the present trend of AMR progression. Antibiotic usage is the major cause of AMR, with more than half of all antibiotics being used inappropriately in people, veterinary practice, and animal husbandry for growth promotion [31]. The persistence of AMR *Salmonella* in poultry flocks has warranted continuous research to identify alternatives to antibiotics. However, this alternative will function differently by avoiding infections, minimizing the evolution of resistance by targeting other mechanisms of action, or boosting the efficacy of current antibiotics. The use of these antibiotic alternatives would eventually decrease both the reliance on antibiotics and the issues associated with AMR *Salmonella* strains. Considerable research has shown that alternatives to antibiotics, such as probiotics, herbal extracts, organic acid, etc., exert variable efficacy in mitigating intestinal *Salmonella* colonization in meat-type poultry [32]. However, prebiotics are often used to modulate the gut microbiota by increasing the development of beneficial bacteria that are already present. Prebiotics have been shown to prevent *Salmonella* outbreaks with negative effects on poultry birds’ gastrointestinal tracts. Prebiotic treatment of meat-type poultry birds enhances gut microbiota alteration, which in turn stimulates the necessary mechanisms to prevent infection by *Salmonella* and horizontal transmission. Meanwhile, postbiotics utilize non-viable bacteria or their products of metabolism, including inactivated organic acids, short-chain fatty acids, enzymes, cells, exopolysaccharides, peptides, and plasmalogens. Lactic acid bacteria are particularly useful owing to their diverse metabolic capacities. Postbiotics in poultry may suppress infections like *Salmonella* and reduce harmful consequences. The oral use of postbiotics has been shown to significantly decrease infections with *Salmonella* in poultry birds. The primary effects can be examined by decreasing *Salmonella* colonization in particular organs, reinforcing the gut microbiota, or stimulating beneficial bacteria such as Lactobacillus. Collateral effects include improved nutrient absorption and growth performance. Postbiotics have been shown to increase production in broilers and laying hens while decreasing infections from *Salmonella* spp. [33]. However, this review is focused on the use of major herbal extracts such as green tea extract, ginger root extract, onion peel extract and their classified bioactive compounds that have shown efficacy in mitigating intestinal *Salmonella* in meat-type poultry.

### 3.2. Extract as an Alternative to Antibiotics for Salmonella Control

#### 3.2.1. Green Tea (GT)

##### Green Tea (GT) Effect on General Health and *Salmonella* Colonization in Broilers

Herbal extracts in the diets of poultry may improve productivity and lower mortality. Green tea (*Camellia sinensis*) is a natural and non-toxic product that contains a wide range of bioactive ingredients, including polyphenols, alkaloids, volatile oils, and polysaccharides [34]. Green tea has a high nutritional value because it contains vital elements such as amino acids, with L-theanine making up over fifty percent of its overall amount of amino acids [34]. Green tea contains polyphenol catechins [35,36]. The leaves include about 82.4% organic matter, 92.2% dry matter, 8.7% ether extract, 19.3% crude fiber, 18.1% crude protein, and 9.8% ash. Notably, green tea has a variety of medicinal qualities, including antioxidant, antibacterial, and immunomodulatory activities [37], making it an important tool in poultry. Green tea reduces cholesterol levels, particularly LDL cholesterol, and lowers the levels of lipoprotein lipase and adipose triglyceride lipase [35]. These impacts may be linked to polyphenols, which promote reverse cholesterol transport by removing cholesterol from peripheral tissues and delivering it to the liver, lowering uptake. El-Deek et al. [38] reported that 1.5 g/kg of green tea significantly reduced plasma triglyceride and total cholesterol levels compared to a control group. The impact of green tea in the dietary intake of broilers subjected to coccidiosis on the way they performed, carcass characteristics, intestinal mucosal morphology, blood constituents, and ceca microflora was investigated by [39], and it was discovered that green tea mitigates lipoprotein lipase and adipose triglyceride lipase [35]. GT impact may be linked to polyphenols, which promote reverse cholesterol transport by removing cholesterol from the peripheral tissues and delivering it directly to the liver thereby lowering uptake. Zhao et al. [40] found that GT leaf meal polyphenols had antimicrobial impacts against fungal, viral, and intestinal bacteria. GT compounds can promote the development of helpful microorganisms while inhibiting infectious ones [41]. The lowering of harmful microorganisms may reduce microbial competition in the gut, allowing good microbes to thrive more freely.

##### Green Tea (GT) Effect on General Health and *Salmonella* Colonization in Turkey

Green tea has received attention for its possible health advantages, including antioxidant properties, anti-inflammatory effects, gut microbiome support, and prebiotic potential. Green tea includes the polyphenol epigallocatechin gallate (EGCG), which functions as an antioxidant. EGCG may neutralize free radicals, possibly lowering the risk of cancer, diabetes, heart disease, and arthritis [42]. Inflammation may disturb the delicate equilibrium of our gut flora, causing havoc in our bodies. However, the polyphenols included in green tea have anti-inflammatory effects. These chemicals may lower inflammation, which is essential for gut health. A new study reveals that green tea may boost gut bacteria. It may even help with leaky gut syndrome, which occurs when the intestinal barrier weakens, letting dangerous substances into the circulation. Maintaining a healthy gut barrier is important, even if leaky gut syndrome is not a medical diagnosis. Some studies suggest that green tea may work as a prebiotic, supporting the development of healthy bacteria in the stomach. Green tea may improve communication between the brain and stomach, which is important for general health. In 2018, there was a multistate epidemic of multidrug-resistant *Salmonella* illnesses associated with raw turkey products. People may contract *Salmonella* by eating undercooked turkey or handling raw turkey, even packaged raw pet food. Turkey should be cooked thoroughly to avoid foodborne illness. Several investigations have looked into the antibacterial effects of green tea components and its variety of health advantages. Green tea seed saponins have been examined for their effectiveness against *Salmonella*. However, [42] further study is required to establish a definitive correlation and it should be noted that green tea may be a beneficial supplement to a healthy lifestyle, but it is critical to follow correct food safety procedures while handling raw poultry meat, such as turkey [42].

##### Green Tea (GT) Effect on General Health and *Salmonella* Colonization in Quail

The use of naturally occurring phytogenics, such as green tea (GT) products, in poultry production is heavily impacted by the desire for antibiotic-free food items. Introducing GT products in poultry species offers significant capacity to improve the quality of meat for customers. GT bioactive substances may also be used to improve the immune status of poultry birds. According to Kara et al. [43], Japanese quail fed diets supplemented with 2.50 g/kg catechin exhibited a notable increase in water-holding capacity and antioxidant capacity, along with reductions in serum glucose and total cholesterol levels in breast meat. However, the antioxidant actions of catechins present in GT might have protected cell walls from lipid peroxidation, which might be a major factor contributing to this development. In contrast to weight gain and feed utilization efficiency, the substitution of the zinc-bacitracin antibiotic with GT leaf powder in the diets of Jumbo quail resulted in a surge in total consumption of feed [44]. This illustrates the potential of GT products to be used as viable substitutes for antibiotics in poultry feeds.

##### Green Tea (GT) Effect on General Health and *Salmonella* Colonization in Ducks

Green tea is recognized for its potential as a source of antioxidants beneficial to health due to compounds such as epigallocatechin gallate (EGCG) and other catechins found in tea leaves [42]. Duck meat, rich in nutrients, faces a high risk of bacterial growth and spoilage, necessitating preservation methods, one of which involves natural green tea preservatives. Green tea contains tannin compounds like catechins, leukoanthocyanin, gallic acid, caffeic acid, and chlorogenic acid, which are phenolic compounds capable of damaging cell wall polypeptides and deactivating host cell molecules, as noted by [45], regarding the antimicrobial properties of phenolic compounds. Previous studies indicate that a 5% concentration of green tea extract can reduce pathogenic bacteria like *Shigella dysenteriae*, *Salmonella* spp., *Escherichia coli*, Staphylococcus aureus, and Listeria monocytogenes [42,45]. Soaking chicken meat in a 5% green tea extract solution in a Petri dish for 10 min increases its shelf life by inhibiting bacteria such as *E. coli*, *S. aureus*, *S. Typhi*, and *Bacillus* [45]. Similarly, treating beef with a 2.5% green tea extract solution for 10 min decreases *S. aureus*, *L. monocytogenes*, *S. typhimurium*, and *E. coli* counts, as well as lipid degradation and color instability [45]. Additionally, flavonoids in green tea can denature proteins, halting bacterial metabolic activity since enzymes, necessary for such activities, are proteins, thereby causing bacterial cell death [45]. These findings are consistent with the idea that soaking chicken meat in a 5% green tea extract solution for 10 min extends its shelf life by inhibiting *E. coli*, *S. aureus*, *S. Typhi*, and *Bacillus* growth. While soaking duck meat in green tea extract significantly impacts its color, it does not notably affect its aroma or taste.

##### Mechanism(s) of Action of Green Tea Extracts

*Camellia sinensis* (L.) is among the oldest and most widely consumed beverages in the world. Green tea is classified primarily based on the tradition of green tea leaf processing, the location of origin, and the kind of soil in which the plant grew. It is cultivated mostly in Japan, China, and Taiwan, and its antioxidant capacity is significantly impacted by the technical process, which results in increased catechin concentration due to the oxidation of catechins to aflavins during fermentation. It contains flavonoids such as (−)-epigallocatechin-3-gallate (EGCG), (−)-epicatechin-3-gallate (ECG), and (−)-epigallocatechin (EGC). Green tea’s chemical makeup contains almost 10 different categories of chemicals. Its primary components are phenolic acids, polyphenolic chemicals (catechins), amino acids, proteins, and fats [46]. Catechins operate as antioxidants by scavenging reactive oxygen compounds, inhibiting free radical production, and preventing lipid peroxidation. Catechins’ antioxidant activity and influence on preventing illness are mostly determined by their structural groups and hydroxyl group count. The polyphenol content, namely flavanols and flavanols, contributes to its beneficial effects on health and clinical investigations, both in vivo and in vitro, and supports their antioxidant and anti-inflammatory properties. The flavonoid content, comprised of catechins, acts as antioxidants by neutralizing free radicals and chelating metal ions during redox processes. The tea contains 15–20% protein, including amino acids like l-theanine [46], tyrosine, and tryptophan, and trace elements like magnesium, chromium, manganese, calcium, copper, zinc, iron, selenium, sodium cobalt, or nickel, as well as carbohydrates like glucose, cellulose, and sucrose [47]. Catechins’ antioxidant activity relies on both their chemical structure and environmental circumstances. Epigallocatechin-3-gallate (EGCG) is the most well studied and abundant catechin derivative and has shown a broad spectrum of actions, including anti-inflammatory, antioxidant, and vasoprotective ones. The catechin content of tea differs by its variety, growing technique, leaf processing, brewing time, and temperature. Catechins serve as chelators for copper and iron ions, and the polyphenol structure, with at least five hydroxyl groups, significantly affects its antioxidant activity. Catechins and other active compounds help repair UVB-induced DNA damage, and have been shown to successfully avoid UV radiation impairment to the epidermis [47].

#### 3.2.2. *Ginger extract* (GE)

*Zingiber officinale* (Ginger) belongs to the Zingiberaceae family and is a semi-woody perennial plant used as a spice and medicinal herb. Ginger rhizomes are mostly composed of carbohydrates (50–70%), lipids (3–8%), terpenes, and phenolic substances [48]. Terpene constituents of ginger contain zingiberene, β-bisabolene, α-farnesene, β-sesquiphellandrene, and α-curcumene, as well as phenolic chemicals such as gingerol, paradols, and shogaol. Gingerols (23–25%) and shogaols (18–25%) are present in greater concentrations

##### *Ginger extract* (GE) Effect on General Health and *Salmonella* Colonization in Broilers

Ginger is used as a medicinal agent and contains active biochemical compounds, such as gingerol, shogaols, gingerdiol, and gingerdione [48]. The supplementation of ginger extract on broiler chickens indicated a substantial increase in performance, but no significant influence on carcass attributes was seen [49]. Ofongo and Ohimain, [50] found that antibacterial properties of fresh ginger root extract administered to broiler chicks resulted in a substantial reduction in the microbial population of gastrointestinal tract’s microbial population, notably *Salmonella* sp., *Lactobacillus* sp., and *E. coli*, a week. Although this action is suggestive that GE can control pathogenic microbial populations, it is recommended that its dosage be carefully considered to prevent eliminating important microbes in the broiler’s digestive tract [50].

##### *Ginger extract* (GE) Effect on General Health and *Salmonella* Colonization in Turkey

Ginger, also known as *Zingiber officinale*, has anti-inflammatory and antibacterial properties [51]. Ginger, as an alternative to growth stimulants made from antibiotics, may boost poultry production, feed palatable qualities, nutrient utilization, malnutrition stimulation, and gastric juice flow [51]. Ginger’s active chemicals (gingerol, shogoals, gingerol, and gingerdione) have been shown to lower triglyceride, LDL, and AST levels. Herbs may reduce LDL cholesterol levels by decreasing the activity of lipogenic enzymes and lowering fatty acid production in the liver. Turkey poults fed a ginger-supplemented diet exhibited higher HDL levels compared to controls. Supplementing chicken diets with natural additions may reduce the risk of elevated blood lipid levels. In an experiment conducted by [52] on the effect of ginger as an herbal extract in turkey poults, the authors found that the treated group had considerably lower total cholesterol, triglyceride, and LDL levels, while HDL cholesterol concentrations rose.

##### *Ginger extract* (GE) Effect on General Health and *Salmonella* Colonization in Quail

Humans have been traditionally using herbal products that have antibacterial properties for ages. In response to antibiotic resistance, research has shifted to natural chemicals as feed additives for cattle. Ginger has shown several biological actions, including antibacterial and antimicrobial capabilities. Dietary supplementation with GP may increase productive performance and egg quality in Japanese quails. A complete investigation is needed to assess the effect of quail dietary ginger supplementation on productivity. Yusuf et al. [53] found that combining ginger with probiotics and organic acid in laying Japanese quail diets increased laying performance, feed conversion ratio (FCR), egg quality, bone properties, and reproductive indicators. Ginger powder at a 0.05 g/kg diet improved egg production, hatching, reproductive performance, and economic efficiency in Japanese quail [53]. The introduction of ginger powder in bird diets has a positive impact on a variety of productivity metrics. Ginger’s active compounds (e.g., brunel, camphon, limonene, humolin, gingerol, gingeron, gingerdiol, shogaols, some phenolic ketone derivatives, volatile oils, alkaloids, saponins, and flavonoids) [53] may stimulate feed digestion and digestive enzymes, increasing FI and FCR [53]. Ginger powder at a 0.05 g/kg diet improved production, hatchability, reproductive performance, and economic efficiency in Japanese quails while also boosting egg weight and feed intake [53]. The effectiveness of ginger on avian performance varies depending on the species, dose, compounds, and interactions with other food components.

##### *Ginger extract* (GE) Effect on General Health and *Salmonella* Colonization in Ducks

Duck diarrhea syndrome, caused by bacterial strains such as *Salmonella*, is one of the illnesses that results in significant losses in duck husbandry. Long-term use of limited antibiotics has resulted in drug resistance, making treatment of duck diarrhea challenging. *Salmonella* spp. was a major contributor to duck diarrhea syndrome. Vietnam has detected a major increase in antibiotic resistance among pathogens such as *E. coli* and *Salmonella*, leading to research into plants with antibacterial qualities to treat animal diarrhea. The hunt for efficient herbal remedies to treat resistant *Salmonella* strains and duck diarrhea is ongoing. Ginger has been widely recognized for its antimicrobial qualities. Ref. [54] found that ginger inhibited multi-antibiotic bacteria, highlighting the benefits of using herbs whenever antibiotic treatments are unsuccessful owing to their resistances. Several antibiotic-resistant bacterial strains were shown to be responsive to medicinal plant components. Herbal extracts such as ginger and its bioactive compounds have antibacterial effectiveness due to their unique method of action, making it difficult for bacteria to acquire resistance to them.

##### Mechanism(s) of Action of Ginger Extracts

*Ginger extract*, a plant with numerous chemical constituents, has been widely used for its potential anti-inflammatory and antioxidant properties. The main components found using the gas chromatography–mass spectrometry (GC-MS) technique include zingiberene (31% of the total area), curcumene (15.4%), and sesquiphellandrene (14.02%). In vitro, ginger extract reduced tissue lipid peroxidation by absorbing radicals such as superoxide, hydroxyl, and DPPH. Ginger oil also increased glutathione reductase, glutathione, and superoxide dismutase levels after at least 30 days of oral administration. The concentration of ginger oil can vary from 1.0 to 3% based on the origin of the rhizomes. Studies have shown that adding herbal additives to neonate chick diets improves broiler chicken immune system response through improving splenic lymphocyte proliferation, antibody titers against pathogens, and promoting good microbial colonization. *Ginger extract* has been found to reduce phospholipids, total cholesterol, triglycerides, and (VLDL or LDL) cholesterol levels in blood and aortic tissue homogenates. Ginger root is abundant in volatile oil compounds, gingerols, and zingerone and is used as herbal remedies worldwide. Zhang et al. [48] discovered that ginger root boosts birds’ digestive enzymes and antioxidant activity. Adding 5000 mg/kg of powdered ginger to broiler meals increases antioxidant capacity and blood metabolites. Quails given a ginger-supplemented diet had the highest feed conversion ratio, body weight, and humoral immunity. Ginger increases the bird’s antioxidant level and optimizes blood serum lipid profiles. Bee propolis and ginger powder have been found to boost chicken growth and health. In conclusion, ginger extract has significant potential for various applications in animal health and reproduction.

#### 3.2.3. Onion Peel Extracts

Onion peel waste has been investigated as a source of nutritional fiber and bioactive substances for livestock and food industries. It also serves as a source of important biomolecules, fertilizers made from organic matter, and a source of renewable energy. Onion peel waste contains chemicals with diverse biological activity that have potential use in the pharmaceutical and food sectors. The outer layers of onions are known to be high in phenolic chemicals, particularly polyphenolic flavonoids, that are more plentiful in the outermost part than the internal and center layers of the bulb. Onion peel has a high quantity of quercetin, most likely owing to sun-induced deglucosidation of its glucosides. Several studies have shown that onion peel has antibacterial characteristics. Onion peel’s antimicrobial compounds have been shown to have proved more effective against Gram-positive bacteria (*B. cereus*, *S. aureus*, *M. luteus*, and *L. monocytogenes*) than Gram-negative bacteria (*E. coli* and *P. aeruginosa*). Onion peel extracts have shown superior antibacterial properties against many harmful bacterium types. Onion peels contain quercetin, which works as an antibacterial agent. It disrupts the metabolism of energy, cell membrane activities, and nucleic acid production. Olugbemi-Adesipe [55] synthesized silver nanoparticles from onion peel and investigated their antibacterial efficacy against *S. typhimurium* (Gram-negative) and *S. aureus* (Gram-positive) bacteria. Both bacteria were effectively inhibited using zones of 9 mm and 8 mm, respectively. Pathogenic bacteria are known to disrupt the functionality of digestive systems in poultry industries, affecting nutritional digestion and absorption. However, the continued use of antibiotics, which has resulted in AMR residues, has led to the search for options from herbal extracts. Onion peel extracts have been shown to boost chicken health, growth, and production. In the poultry industry, pathogenic bacteria disrupt the functionality of the digestive systems which actually affect the digestion and absorption of nutrient uptakes. However, continued use of antibiotics which has resulted in AMR residues has prompted to seek alternatives such as herbal extracts. onion peel extracts have been evaluated to improve chicken health and performance of growth as well as their productivity. Olugbemi-Adesipe [55] indicated that onion peel extract contains various phytochemical compounds that mitigate against pathogenic bacterial population as well as enhanced digestion and absorption of nutrients without affecting the feed intake and total weight gain. The author deduced that onion peel extract is a valuable herbal extract that can be used in poultry industries without compromising the attributes and carcass yield of poultry meat. It was noted that since onion extracts had no negative effect on the productivity of chickens, and hence could be used as an antibiotic’s alternative and as growth promoter in poultry production. However, despite the effectiveness of this herbal extract, it has not been effectively tested in other poultry birds.

#### 3.2.4. Guava Leaf Extract

Guava (*P. guajava*) is a tropical herbal extract that has been used to treat various gastrointestinal disorders, including diarrhea and gastric pain. Its extracts have anti-inflammatory and hemostatic properties. Phytochemical investigations have identified over twenty compounds in guava extracts, with tannins, P-sitosterol, being the most important elements. Guava leaf extract has excellent antibacterial effects against pathogens like *S. aureus*, *E. coli*, *P. aeruginosa*, *B. subtilis*, and *S. typhimurium*. *P. guajava* leaves contain several bioactive components, including glycosides, phenol, flavonoids, terpenoids as well as saponins. Research suggests that bioactive chemicals found in guava leaves work as antibacterial agents. Several authors have documented the presence of phytochemical substances in *P. guajava* extracts as being accountable for antibacterial action [56], notably alkaloids and tannins, which have been extensively studied for their effectiveness against bacteria. The herbal extracts effectively inhibited the bacteria, with aqueous extracts exhibiting greater inhibition zones. Products from herbal extracts are known to contain effective chemical compounds against pathogenic bacteria. Research found that guava extracts, an organic compound, may enhance the reliability and overall quality of many poultry products. Guava leaf extracts, at a concentration of 5%, effectively suppress the development of *Salmonella*, suggesting its potential as a natural component for pathogen management. However, this herbal extract has not been widely studied and used in meat-type poultry production either as growth promoter or as an alternative to antibiotic in reducing the impact of AMR. More research needs to be carried out in order to ascertain their effective utilization [56].

#### 3.2.5. Essential Oils

Essential oils are volatile, non-toxic, aromatic substances that are naturally occurring and derived from a variety of plant components. They positively impact the performance of broilers as effective growth promoters, immunostimulants, antioxidants, antibacterial, antifungal, and antiparasitic compounds. The extract of lemongrass (*Cymbopogon citratus*) exhibits notable antibacterial properties against several pathogenic microorganisms, such as *S. enterica* and *S. typhimurium*. Moreover, the administration of essential oils extracted from oregano and thyme effectively impeded the colonization of *Salmonella* species within the gastrointestinal tract of chickens. Additionally, it has been reported that carvacrol, thymol, trans-cinnamaldehyde, and eugenol possess antibacterial properties against *Salmonella* and *Campylobacter* in broiler poultry [57]. Cinnamaldehyde, an aldehyde discovered in the bark of cinnamon plants, imparts the unique fragrance of cinnamon which serves as antifungal, antibacterial, and anti-inflammatory [57]. Cinnamaldehyde exhibits a wide range of antibacterial activities through several mechanisms, including alteration of membrane permeability and inhibition of glucose consumption [57]. In a study on the antibacterial properties of cinnamaldehyde observed against *S. enteritidis*. [57], it was found that the application of 10 mM cinnamaldehyde reduced the proliferation of *S. enteritidis*. Cinnamaldehyde has been utilized in the diet of chickens to prevent gastrointestinal illnesses, and it was found that the supplementing broiler chicks daily with cinnamaldehyde at concentrations of 0.5% or 0.75% resulted in a significant reduction in the count of cecal *S. enteritidis* [57]. However, the use of essential oil has not been used in other poultry species in reducing intestinal *Salmonella* colonization.

### 3.3. Mechanism(s) of Action of These Additives

Phenolic chemicals are bioactive secondary metabolites found in medicinal plants, often used to treat pathogenic microorganisms [57]. Flavones, flavonoids, and sulfur-containing compounds have been extensively studied for their antibacterial, antifungal, antiviral, and antiprotozoal properties. Flavonoids disrupt microbial envelopes and form complexes with microorganism cell walls, inactivating certain microbial enzymes. Flavonoids have shown significant action against *E. coli* and *Mycobacterium* TB. Sulfur-containing compounds derived from plants with high polysulfides, such as allicin, ajoene, and isothiocyanates, have shown efficacy against both Gram-positive and Gram-negative bacteria, including Helicobacter pylori. Coumarins are a type of phenolic substance that can kill germs in both natural and synthetic states. Coumarins, which are present in medicinal plants, can kill a variety of different bacteria. These bacteria include *Bacillus subtilis*, *Salmonella typhi*, and *Staphylococcus aureus*. Terpenes, the primary components of essential oil fractions, have been found to enhance antibacterial properties by collaborating with other active chemicals. Pathogens like *E. coli*, *S. aureus*, and *Salmonella* spp. have been shown to be inactivated by these chemicals [57]. The activities of these bioactive compounds and herbal plants against selected pathogenic strains of *Salmonella* is presented in Table 1 and Table 2.

## 4. Potential Development of Resistance or Cross-Resistance by *Salmonella* to These Additives

With the growing interest and acknowledgment of plant-derived compounds’ antimicrobial capabilities against *Salmonella*, it is imperative to delve into the potential for resistance or cross-resistance development within *Salmonella* strains. The key is understanding resistance mechanisms to ensure these natural remedies containing bioactive compounds remain effective. This review addresses resistance development, causes, and mitigation strategies given the ability of *Salmonella*’s to adapt, including genetic mutations and acquiring resistance genes, investigating genetic factors behind resistance leading to reduced susceptibility to plant-derived compounds, which is crucial. However, exposure to low or sub-lethal dosages of plant-derived compounds can create selective pressure on *Salmonella* populations, encouraging the growth and spread of resistant strains as a means of species adaptation. Studies have been directly focused on determining minimum inhibitory concentrations (MICs) of bioactive plants to prevent pathogenic resistance and establish recommended controlled usage protocols. Cross-resistance, where resistance to one compound may imply resistance to others, poses a challenge, making it important to understand cross-resistance patterns for effective resistance management. This is a new trajectory in the field of ethno-medicine which requires thorough investigation, innovative research methodologies, and interdisciplinary collaboration to fully understand and harness the potential of traditional remedies with the numerous beneficial bioactive in poultry health management.

## 5. Potential Synergistic Effects of Combining Multiple Herbal Extracts and Bioactive Compounds

Several species of herbal extract discovered to have therapeutic properties have been used for years in traditional treatments for illnesses or syndromes across the globe. Herbal extract including bioactive compounds such as alkaloids, flavonoids, phenolic compounds, steroids, tannins, terpenoids, and other secondary metabolites have powerful antiparasitic and antipathogenic effects. Herbal extract derived compounds have distinct pharmacological features, including low cost, low toxicity, fewer side effects, and a lower likelihood of developing resistance. Combining different herbal extracts may enhance pharmacologic efficacy by achieving synergism, targeting numerous targets concurrently, decreasing dosages, and minimizing adverse effects. To optimize the chemical properties of herbal extracts, it is crucial to identify which of their potential effects leads to the mixed effect. Determining the reactions that occur between the active substances in herbal plant extracts is crucial, despite the complicated and variable nature of the search. Herbal extracts use a varied set of active chemicals to target microorganisms. The combined impact might differ depending on the targeted species of bacteria [92]. Identification of the biologically active substances responsible for specific biological activities in medicinal plant extracts may be challenging due to their abundance [5]. Herbal extracts work by combining various chemicals with synergistic, additive, or antagonistic effects to provide overall efficacy [12]. Further study is needed to determine the impact of synergy, increased bioavailability, cumulative effects, or additive characteristics. The therapeutic effectiveness of herbal extracts relies on synergistic interactions between individual or combination components. To ensure effectiveness, herbal plant extract synergy should be rigorously analyzed and verified in clinical studies. Furthermore, the bioactive molecules responsible for these effects and their interactions are still not well known [79]. Multiple bioactive chemicals may interact in three ways: synergistic, additive, or antagonistic. Synergy occurs when the combined impact of many chemicals exceeds their separate effects. Synergy occurs when one chemical increases the therapeutic action of another by modulating its absorption, distribution, metabolism, and excretion. Additionally, synergy may occur when inert substances become active when combined. Herbal plant synergy is enhanced by the plant’s biochemical matrix, which contains a diverse range of substances, rather than individual compounds’ cumulative effects. This synergy alters biochemical processes, affecting membrane potential, receptor specificity, and protein changes. The originality of herbal plants stems from their usage in mixtures and the interactions of bioactive components. Synergy, a significant component in medicinal plant treatment, occurs when a combination of substances produces larger results than their individual contributions [48]. Using several antimicrobial agents may have varying effects based on their composition and concentration. Synergy occurs when two antimicrobial chemicals work together to create greater antibacterial effects than each of the compounds alone.

## 6. Problems Associated with Developing New Antimicrobials from Herbal Extracts

Indigenous traditional knowledge about medicinal plants holds promise for the development of biocompatible remedies and the identification of novel antimicrobial agents. The use of medicinal plants as antibacterial agents faces challenges due to the lack of standardized treatment protocols and the challenge of understanding the structure–activity connection and mechanisms of action of bioactive substances [93]. Investigation into the pharmacology of medicinal plants is crucial for establishing standardized therapy regimens. The repeatability of plant extract composition is a significant constraint, as accurate identification and verification of bioactive substances are necessary for quality control processes [94]. Standardization of extraction methods and in vitro testing procedures is recommended to enhance systematic exploration of medicinal plants for new antimicrobial drugs and aid in accurate research interpretation [95]. Despite the growing abundance of molecules derived from antimicrobial medicinal plants, the use of plant-derived pharmaceuticals in clinical settings remains limited due to the intricate interactions among their components. The collective efficacy of plant extracts may be attributed to combinations of chemicals exhibiting synergistic, additive, and antagonistic effects. The categorization of combination effects in intricate mixes and the determination of contributing compounds present a challenge, especially with existing methodologies focusing on simplifying complexity and identifying individual active compounds in mixtures of natural products. Metabolomics developments are revolutionizing the identification and effective application of naturally occurring antimicrobials. Statistical modeling has been used to predict and correlate the metabolomic profile and bioactivity of extracts, but there is a lack of consensus on the most effective reference models for defining combination effects. Newer models, such as the zero-interaction potency model and the specific mean equation, may help identify combination effects. Specifically, hydrogel formulations and active packaging materials, as well as emerging technologies such as bio-adhesive technology and nanotechnology, possess the capacity to augment the effectiveness of plant antimicrobial compounds. However, precise simulation of digestive parameters in vivo remains unattainable. The examination of the toxicity of medicinal plant extracts as antimicrobials is a significant barrier. The Food and Drug Administration in the United States has not addressed the majority of extracts, resulting in a dearth of authoritative information on the real toxicity of many extracts. Improper labeling, standardization, and insufficient identification and authenticity can contribute to plant extract toxicity. Extract regulation necessitates stringent production standards as well as regulatory control. Inadequate financial backing for research hinders high-quality investigations into the structure–activity relationship of specific compounds. Despite these challenges, the pursuit of novel antibiotics derived from traditional medicinal plants remains in high demand [96].

## 7. Research Gaps

The potential synergistic or antagonistic effects of bioactive compounds within the herbal extracts, as well as limited research studies for assessing the efficacy and potential side effects of using herbal extracts in treating transmissible diseases, are part of many research gaps that need to be addressed for effective utilization of herbal extracts [97]. However, standardized experiments are required to assess the antibacterial effectiveness of purified bioactive compounds, whereas herbal extracts or plant essential oils need specialized experimental approaches. Another key restriction is the repeatability of herbal extracts’ composition. It is recognized that the same extract might have varied qualities depending on the quality and supply. Establishing quality control processes requires accurate characterization and verification of bioactive substances. Quality plant species are confined to a certain geographic location. Various factors, including the species of plant and the environment, might impact the variety of medicinal herbal extracts’ availability [97]. More research should be carried out on the use of herbal extract that can serve as an alternative to antibiotics in combating *Salmonella* colonization in other meat-type poultry birds such as turkey, duck, pheasant, and quail, which are less studied in poultry production.

## 8. Future Directions

*Salmonella* contamination in poultry remains a critical concern, leading to economic losses and jeopardizing public health. Conventional antimicrobials have limitations such as antimicrobial resistance and environmental impact, prompting the exploration of herbal extracts as potential alternatives. These extracts derived from various plant sources exhibits antimicrobial properties and possess the potential to revolutionize *Salmonella* control strategies in poultry birds. Several herbal extracts and their bioactive compounds have demonstrated antimicrobial activities against *Salmonella* spp. However, their efficacy can be influenced by factors such as extraction methods, formulation, and concentration. To optimize their full potential, future research should focus on refining extraction techniques, standardizing formulations, and determining the ideal concentration for maximum efficacy. One promising avenue for future research involves exploring synergistic combinations of these herbal extracts. However, combining different herbal extracts with complementary antimicrobial properties may result in a more robust and effective control strategy against *Salmonella* spp. in meat-type poultry products. Understanding the synergistic interactions between these compounds could lead to the development of novel formulations with enhanced efficacy, reducing the reliance on single extracts. Elucidating the mechanisms of action underlying the antimicrobial properties of herbal extracts is crucial for their successful implementation in poultry, such as broiler chicken production. Research should focus on unraveling how these extracts and their compounds disrupt *Salmonella* growth, interfere with biofilm formation, and modulate bacterial virulence factors. This knowledge will not only enhance our understanding of the antimicrobial mechanisms but also guide the development of targeted strategies to combat *Salmonella* effectively. The development of effective delivery systems is another critical aspect of optimizing herbal extracts for *Salmonella* control. Researchers should explore innovative delivery mechanisms to improve the bioavailability and stability of these extracts within the meat-type poultry birds’ gastrointestinal tract. Nanoencapsulation and microencapsulation technologies, for instance, could enhance the controlled release of active compounds, ensuring sustained antimicrobial effects. As the poultry industry seeks sustainable practices, it is essential to assess the environmental impact of herbal extracts. Future research should focus on evaluating the ecological footprint of these biomaterials and their by-products, ensuring that their utilization aligns with environmentally responsible practices. Sustainable production methods and waste utilization strategies should be explored to maximize the eco-friendly aspects of plant-derived solutions.

## 9. Conclusions

Several mitigating strategies have been employed in reducing the effect of this pathogenic bacteria, including the use of antibiotics, but the issue of antibiotic resistance and recent government ban has limited their usage. There has been a need to develop and verify viable alternatives of natural origin that are both accessible and economically advantageous in order to effectively manage *Salmonella* infection, prevent disease, and enhance the overall productivity of avian species. Plant extracts have been shown to contain several antimicrobial activities, which signifies new expectations for preventing the increasing antimicrobial resistance threats. The identification of bioactive compounds from herbal extracts is of great importance, showing promising results in the management and control of *Salmonella* in meat-type poultry birds. Phytobiotic feed additives have shown promising results in controlling *Salmonella* infection and improving poultry output due to their pharmacological properties, such as stimulating consumption and enhancing antioxidant properties. However, optimizing plant-derived biomaterials for controlling *Salmonella* in meat-type poultry birds represents a promising frontier in poultry production. Future research directions should encompass refining extraction techniques, exploring synergistic combinations, elucidating mechanisms of action, developing effective delivery systems, and assessing environmental sustainability. By addressing these challenges, the poultry industry can harness the full potential of herbal extracts and their bioactive compounds, ensuring safer and more sustainable meat-type poultry birds production practices in the years to come.

## Figures and Tables

**Figure 1 animals-14-01087-f001:**
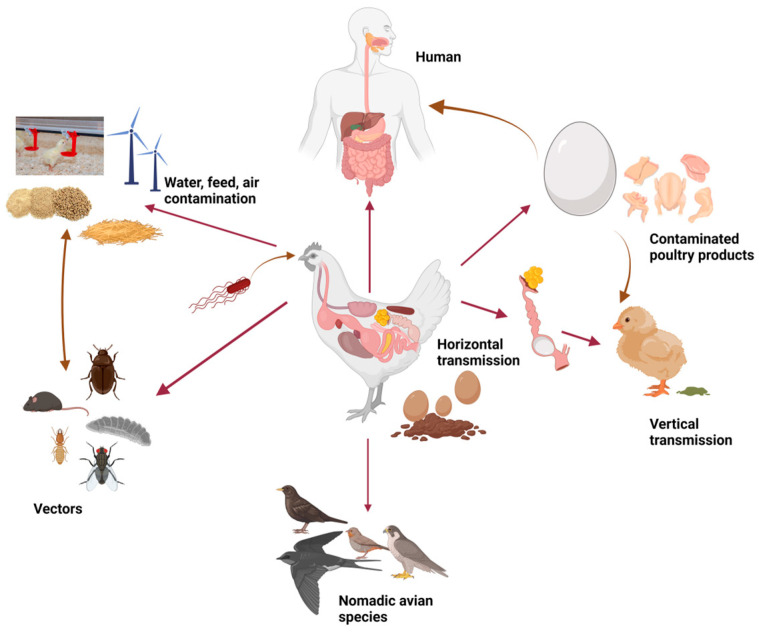
Summary of the different pathways through which *Salmonella* is transmitted.

**Table 1 animals-14-01087-t001:** Classified bio-active compounds in herbal extracts against *Salmonella*.

Compound	Class	Plant Found	Pathogenic Strain	Reference
Myricetin	Flavonoids	Tea, berries, mangifera, onions and herbs	*S. enteritidis*, S. *Paratyph*, *Salmonella* typhi CMCC 50013	[58,59]
Thymoquinone	Pan-assay interference compound	Black cumin seed, Monarda fistulosa plants	S. *Paratyph*, S. *cholerasuis* subsp. *cholerasuis* ATCC 10708, *S. enteritidis*	[60]
Rutin	Pan-assay interference compound	Black tea, green tea, eucalyptus, tobacco, forsythia, hydrangea, and viola	*S. paratyph*, *S. cholerasuis* subsp. *cholerasuis* ATCC 10708, *S. enteritidis*	[59,61,62]
Myricetol	Flavonol glycoside	Bambara groundnut, strawberries, and spinach	*S. paratyph*, *S. cholerasuis* subsp. *cholerasuis* ATCC 10708, *S. enteritidis*	[63,64,65,66]
Nerolidol	Flavonoid	Ginger, jasmine, lavender, and lemon grass	*S. paratyph*, *S. cholerasuis* subsp. *cholerasuis* ATCC 10708, *S. enteritidis*	[67,68]
Isophytol + Ginkgo biloba polyprenols	Sesquiterpene alcohol	Red algae, chamomile, and Ginkgo biloba leaves	*Salmonella enterica*	[67]
Morin	Flavonoid	Guava, onions, apple skins, and figs	*Salmonella* Enteritidis	[67,68,69,70]
Anthocyanins	Phenolic	Elderberries, black carrots, red cabbage, purple cauliflower, red potato tuber, purple potato tuber, and purple pepper fruit	*S. typhimurium*	[71,72,73]
Catechins	Flavonoid	Apricots, guava leaves, and fresh tea leaves	*Salmonella*	[74]

**Table 2 animals-14-01087-t002:** Different herbal extracts, their botanical name and extraction methods.

Common Name	Botanical Name	Part Used	Solvent	Pathogenic Strain	Reference
Aloe vera	*Aloe barbadensis*	Leaf	Methanol	*Salmonella enterica*	[75]
Cinnamon	*Cinnamomum verum*	Leaf	Aqueous	*Salmonella* Typhimurium	[76]
Turmeric	*Curcuma longa*	Rhizomes	Chloroform	*S. typhimurium*	[77]
Bael tree	*Aegle marmelos*	Fruits	-	*S. typhimurium*	[78]
Spotted Pumpkin	*Lagenaria breviflora*	Fruit	Methanol	*Salmonella* spp.	[79]
Guinea hen weed	*Petiveria alliacea*	Leaves	Methanol	*Salmonella*	[79]
black pepper	*Piper nigrum*	Seeds	Aqueous	*S. typhimurium*	[80,81]
Guava	*Psidium guajava*	Leaves	Acetone	*S. typhimurium*	[82]
Cinnamon	*Cinnamomum zeylanicum*	Dried powder	Methanol	*S. typhimurium*	[83]
Cinnamon	*Cinnamomum zeylanicum*	Essential oil	-	*S. typhimurium*	[84]
Lemongrass	*Cymbopogon citratus*	Essential oil	-	*S. typhimurium*	[85]
Blackberry	*Blackberry (Rubus fruticosus)*	Pomace extract	-	*S. typhimurium*	[86]
Blueberry	*Blueberry (Vaccinium corymbosum)*	Pomace extract	-	*S. typhimurium*	[86]
Greek sage	*Salvia fruticosa*	Leaves	Methanol	*S. typhimurium*	[61]
Stone breaker	*Phyllanthus amarus*	Leaves	Ethanolic	*S. typhimurium*	[87]
bush tea	*Athrixia phylicoides*	Leaves	Ethanol	*S. typhimurium*	[88]
special tea	*Monsonia burkeana*	Leaves	Ethanol	*S. typhimurium*	[88]
Cranberry	*Vaccinium macrocarpon*	Fruits	Ethanolic	*Salmonella enterica serovars* Typhimurium, Enteritidis, and *Heidelberg*	[89]
Apricot	*Mimusopsis comersonii*	Pulp and seed extracts	Phenolic	*Salmonella* Typhimurium	[65]
Cambess	*Kalanchoe brasiliensis*	Leaves	Ethanol	*Salmonella*	[90]
Guava	*Psidium guajava*	Leaves	Ethanol	*Salmonella* Enteritidis	[91]
Sage	*Salvia officinalis*	Leaves	Ethanol	*Salmonella* Enteritidis	[91]
Rhamnus	*Ziziphusspina christi*	Leaves	Ethanol	*Salmonella* Enteritidis	[91]
Arjun tree	*Terminalia arjuna*	Bark and leaves	N-butanolic	*Salmonella* Typhimurium	[92]

## Data Availability

Not applicable.

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
