# Peer review of "Mitigation Potential of Herbal Extracts and Constituent Bioactive Compounds on Salmonella in Meat-Type Poultry"

_animals, 2024, doi:10.3390/ani14071087_

Round 1

Reviewer 1 Report

Comments and Suggestions for Authors

The aim of this paper was to revise the effects of herbal extracts and bioactive compounds on Salmonella spp. and evaluate their potential to replace antibiotics in meat-type poultry.

The review first accurately analyses the prevalence and effects of Salmonella in poultry farms, its impact on human health and the diffusion of the antibiotic resistance phenomenon, then investigates the potential of herbal alternatives to antibiotics such as green tea, ginger, onion peel and guava leaf extracts and essential oils in different species of poultry birds, specifically chicken, turkey, quail and duck.

However, there are some issues that need to be addressed before publication:

- Please, check the partition in paragraphs and sub-paragraphs. Format the paragraph title in bold and the sub-paragraphs ones in italics, avoiding the presence of letters before the sub-paragraphs’ titles. It would be better to point out the number of the main paragraph to which the sub-paragraph belongs to (eg: 2.1, 2.2, 2.2.1 ecc…).

- The paragraph “Factors influencing Salmonella colonization in meat-type poultry birds” and its sub-paragraphs are sometimes too detailed; please thin them out focusing just on information that may be useful to develop the review.

- Sometimes the paragraph title doesn’t correspond to the paragraph content, e.g. the paragraph entitled “Gastroenteritis as foodborne illness and its general symptom in humans” is about gastroenteritis in poultry. Please revise the title or the paragraph content.

- Please, avoid the repetitions. E.g. in the paragraph “Mechanism(s) of action of green tea extracts” the word “Green tea” is repeated more than 10 times.

- Please remember to always format the bacterial genus and species in italics.

- Lines 17-19: please with “Salmonellosis, a systemic infection caused by Salmonella, an intracellular bacterium able to cause systemic infections…”.

- Line 20: please replace “antibiotics resistance bacteria” with “antibiotic resistant bacteria”.

- Lines 21-24: please replace with “Both in vivo usage and in vitro studies have demonstrated the effective antimicrobial activities of bioactive compounds from herbal extracts against pathogenic bacteria, showing promise in managing Salmonella infections and enhancing poultry performance, so this review delves into control of Salmonellosis using these compounds.”

- Lines 93-95: please replace with “However, the spread of antibiotic resistance genes has necessitated the need to look for alternative to antibiotic usage in meat type poultry production.”

- References: please verify that each reference corresponds to the journal standards.  

Comments on the Quality of English Language

Few minor typos are present and should be addressed before manuscript publication.

Reviewer 2 Report

Comments and Suggestions for Authors

The review article by Orimaye et al. covers the effect of herbal extracts and bioactive compounds effect on the mitigation of Salmonella occurrence in meat-type poultry. The topic is novel and interesting. The article is very informative and nicely written without any major language faults. I have the following observations in this regard as-

A.    The authors also focus on antimicrobial resistance throughout the manuscript and use plant extracts as an alternative to control Salmonella. Thus, authors may add this to the title to increase visibility and widen the readership of the article.

B.     Tables need to be more informative.

C.     Some sections prior to Chapter 3 may be concise, as the main focus of the review is the use of plant extracts, and it starts at Chapter 3.

Other observations as-

        i.            Title: Authors may add Salmonella prevalence or occurrence in the title. Also may remove bioactive component

      ii.            Abstract: L21-27: Please include more specific recommendations and conclusion

    iii.            Keywords: The article may delete pathogens and add bioactive compounds, as it focuses on Salmonella, which has been covered.

    iv.            L57-58: Please cite suitable references for the claim

      v.            L321: the section may be shifted to the above sections

    vi.            Fig 1: Excellent and very informative

  vii.            Table 1: please add more information as the diet, days of supplementation, etc.

viii.            Table 2: please correct pathogenic strain duplicate column. May be solvent

    ix.            Also, in table 2, please add more information on the extraction protocols

      x.            Research gap and future direction: appropriate

Some minor issues can easily be addressed during revisions, are

    xi.            L66: Formatting of reference

  xii.            Please check the manuscript for scientific names to be italics

xiii.            Proper demarcation between sub-chapters 

Comments on the Quality of English Language

Minor issues

Reviewer 3 Report

Comments and Suggestions for Authors

Articles describing the beneficial effects of various bioactive compounds on meat quality are important for understanding their potential role as substitutes for antibiotics. The authors prepared an interesting article describing the mechanism of action of herbal extracts in controlling Salmonella in poultry production.

However, the article has many  points that should be improved:

1. throughout the article, there are numerous errors in the written names of bacteria and extracts, which when given in Latin should be written in italics (ex. lines 360, 502, 503, 505, 507, 512, 517, 562, 655 and many others)

2. in the whole article authors gave incorrect spelling of the name Salmonella Typhimurium and many others, in international nomenclature names of genera are given in italics, and the names of serovars are in big letters without italics (ex. lines 76, 80, 338, 343, and many others)

3. line 142 - why "Polutry" was given in a big letter?

4. Table 1 - S. Paratyphi not Paratyph (in many places in the table)

5. Table 2 - different forms of spelling of genus and species names, standardize these discrepancies

6. line 505 - there is no S. Typhoid, the only described serovar is S. Typhi responsible for typhoid

7. line 504 - on which stage this soaking takes place, please describe

8. line 581 - What study? change the construction of the sentence; after the bracket small letter, why in one sentence the same reference was given twice?

9. line 66 - the number of references, not year of publication

10. line 40 - the sentence is incomprehensible, where is the verb in this sentence?

11. the article requires a change in the layout and a clearer division into sections, the use of letters such as a, b, i, ii is difficult to trace the relationships

besides, authors sometimes use a paragraph for subpoints of the same importance, sometimes not (ex. lines 452 and 478, or 647 and 684)
